# Evaluation of Nitrosative/Oxidative Stress and Inflammation in Heart Failure with Preserved and Reduced Ejection Fraction

**DOI:** 10.3390/ijms242115944

**Published:** 2023-11-03

**Authors:** Karol Momot, Kamil Krauz, Katarzyna Czarzasta, Maciej Zarębiński, Liana Puchalska, Małgorzata Wojciechowska

**Affiliations:** 1Chair and Department of Experimental and Clinical Physiology, Laboratory of Centre for Preclinical Research, Medical University of Warsaw, 02-097 Warsaw, Poland; karolmomot@icloud.com (K.M.); kamilkrauz01@gmail.com (K.K.); katarzyna.czarzasta@wum.edu.pl (K.C.); liana.puchalska@wum.edu.pl (L.P.); 2Department of Invasive Cardiology, Independent Public Specialist Western Hospital John Paul II, Lazarski University, 05-825 Grodzisk Mazowiecki, Poland; maciej@zarebinski.pl

**Keywords:** heart failure pathogenesis, nitric oxide, nitrosative stress, oxidative stress, inflammation

## Abstract

Heart failure (HF) is a complex syndrome characterized by impaired cardiac function. Two common subtypes of HF include heart failure with preserved ejection fraction (HFpEF) and heart failure with reduced ejection fraction (HFrEF). In this study, we aimed to evaluate and compare the plasma levels of 3-nitrotyrosine (3-NT)—as a marker of nitrosative/oxidative stress and myeloperoxidase (MPO)—as an indicator of inflammation between HFpEF and HFrEF. Twenty-seven patients diagnosed with HFpEF and twenty-two with HFrEF were enrolled in this study. Additionally, forty-one patients were recruited for the control group. An echocardiographic assessment was conducted, followed by the collection of blood samples from all participants. Subsequently, the levels of 3-NT and MPO were quantified using the ELISA method. Comprehensive clinical characteristics and medical histories were obtained. Circulating levels of 3-NT were significantly higher in the HFpEF patients than in the control and the HFrEF groups. Nitrosative/oxidative stress is significantly intensified in HFpEF but not in HFrEF.

## 1. Introduction

Heart failure (HF) remains a severe problem for individual patients and public health. Approximately 1–20 new cases are diagnosed per 1000 persons every year [1]. In 2017, around 64.3 million people worldwide had HF [2]. However, its prevalence depends on the geographical area. HF occurs more often in older adults and is related to high mortality in this group [3]. About half of all cases of HF constitute HF with preserved ejection fraction (HFpEF) [4].

HFpEF and heart failure with reduced ejection fraction (HFrEF) have different pathogeneses. Nevertheless, chronic inflammation plays a crucial role in both types of HF. In HFpEF, the systemic and cardiac inflammation results from metabolic risk factors (type 2 diabetes, obesity, and hypertension) and causes the activation of the endothelium in myocardial microcirculation, leading to the upregulation of NADPH oxidase 2 (NOX2) [5]. This results in oxidative stress, increased levels of H_2_O_2_, the uncoupling of endothelial nitric oxide synthase (eNOS), and the decreased production of nitric oxide (NO). In cardiomyocytes, diminished NO bioavailability leads to the lower stimulation of soluble guanylate cyclase (sGC), reduced formation of cyclic guanosine monophosphate (cGMP), and reduced protein kinase G (PKG) activity. The latter is associated with decreased titin phosphorylation and increased passive stiffness of cardiomyocytes [6]. The inflammatory state in HFrEF may be triggered by myocardial infarction (MI) [7]. This inflammation is sustained throughout heart tissue regeneration, remodeling, and scar formation, and is associated with the infiltration of neutrophils, followed by monocytes and macrophages [8]. In this process, the renin-angiotensin-aldosterone system plays a significant role [9].

Recently, Schiattarella et al. established the critical role of nitrosative stress in the pathogenesis of HFpEF. The enhanced activity of inducible nitric oxide synthase leads to the improper functioning of the Inositol-Requiring Enzyme 1α - X-Box Binding Protein 1 (IRE1α-XBP1) pathway, playing a role in unfolded protein response [10]. 3-nitrotyrosine (3-NT) is a marker of nitro/oxidative stress. It is produced from tyrosine in the presence of reactive nitrogen species. Nitration can occur for free tyrosine and amino acids in the polypeptide chain [11]. Loek van Heerebeek et al. discovered that 3-NT expression was elevated in myocardial homogenates from patients with HFpEF compared to HFrEF. However, no studies have been conducted to assess and compare the circulating levels of 3-NT in patients with HFpEF and HFrEF or healthy individuals [12].

Myeloperoxidase (MPO) is a heme protein derived from leukocytes, which is related to vascular dysfunction and is a prognostic factor in cardiovascular pathologies [13]. It has been implicated in the pathogenesis of several inflammatory conditions, including HFpEF [14]. However, some authors have demonstrated a strong relationship between higher levels of MPO and the incidence of HFrEF but not HFpEF [15]. Increased circulating MPO plasma concentrations are associated with increased fibrosis and structural remodeling of the myocardium [16]. Under oxidative stress, human endothelial cells express MPO, which consumes NO and converts it to a nitrogen radical, causing protein nitration and tissue damage [17]. This also results in the reduction in NO levels and correlates with microvascular dysfunction [18]. NO deficiency causes the dysregulation of the NO-sGC-cGMP signaling pathway, which plays a role in development of endothelial dysfunction and hypertension. Decreased cGMP concentrations lead to the inadequate activation of its signaling targets and contribute to aberrant vascular toning [19,20]. 

## 2. Results

The baseline characteristics of patients are presented in Table 1. Echocardiographic features are displayed in Table 2. The results of the measurements for 3-NT, MPO, and NT-proBNP are depicted in Figure 1. 

The linear and rank correlation charts are presented in Figure 2. Regardless of the group, age negatively correlates with Hemoglobin (Hgb) levels (r = −0.28, *p* = 0.0081) and estimated Glomerular Filtration Rate (eGFR) (r = −0.36, *p* = 0.0006). The control group had no correlations between MPO and 3-NT and any other variables. 

Obese patients have greater MPO levels than the non-obese patients (0.91 (0.45–1.41) vs. 0.64 (0.39–0.97), AU/mL, *p* = 0.049). Patients with thyroid disease present higher 3-NT concentrations compared to healthy subjects (9.42 (7.55–12.14) vs. 14.22 (10.09–22.05), ng/mL, *p* = 0.008). Patients with renal dysfunction present higher NT-proBNP levels than those with normal eGFR (270 (172.5–1014) vs. 159 (77–267), pg/mL, *p* = 0.0123). 

CAD and history of MI are the states related to higher NT-proBNP levels (242 (157–794) and 273 (154–863), pg/mL, respectively) when compared to the healthy state (120.5 (73.5–234.75) and 163 (77–268), pg/mL, respectively) (*p* = 0.0016 and *p* = 0.0311, respectively). Regardless of the group, therapy with Sodium-Glucose Co-Transporter 2 Inhibitors (SGLT2i), Angiotensin-Converting Enzyme Inhibitors (ACEI), Angiotensin II Receptor Blockers (ARB), Statins, Beta-blockers, or non-vitamin K antagonist oral anticoagulants (NOAC) does not significantly impact levels of MPO and 3-NT.

## 3. Discussion

In this study, circulating levels of MPO and 3-NT as markers of inflammation and oxidative/nitrosative stress, respectively, were assessed in patients with HFpEF and HFrEF and healthy controls. Our results revealed that HFpEF is associated with significantly increased plasma concentration of 3-NT compared to other groups, suggesting that the pathogenesis of HFpEF is associated with oxidative/nitrosative stress. Using quantitative immunohistochemistry, van Heerebeek et al. showed that myocardial 3-NT levels were significantly higher in samples obtained from patients with HFpEF than HFrEF, which aligns with our results [12]. Kolijn et al. observed higher concentrations of 3-NT in the left ventricular myocardium from HFpEF patients than from healthy donors. Moreover, the skinned fibers obtained from the biopsies were treated in vivo with empagliflozin. This resulted in a significant reduction in 3-NT levels, suggesting that the medication attenuated nitrosative stress [21]. 

Shishehbor et al. revealed significantly increased levels of 3-NT in patients with coronary artery disease compared to healthy controls [22]. Moreover, statin therapy significantly reduced circulating levels of 3-NT, which suggests that it could also be beneficial in HFpEF [23,24]. In the rat model of diabetic cardiomyopathy, treatment with irisin significantly reduces cardiac 3-NT levels [25]. However, no other research has explored irisin’s impact on plasma 3-NT levels in the HFpEF model.

Within the scope of our investigation, the use of medications such as NOACs, statins, and SGLT2 inhibitors did not exhibit a significant correlation with the levels of 3-NT and MPO. However, it is important to note that investigating the impact of these medications on these markers’ levels was not our study’s primary objective. To assess the influence of therapy with these drugs on nitrosative/oxidative stress or inflammation, it would be necessary to measure these markers’ levels before initiating treatment.

In our study, MPO concentrations do not differ between the three groups, which is contrary to the study of Hage et al., who described significantly higher levels of MPO in HFpEF patients compared to healthy controls. Another study, which compared 33 biomarkers between HFpEF and HFrEF, showed that circulating levels of MPO were similar in both groups [14]. Similar to our results, body mass index (BMI) positively correlated with MPO among the HFpEF population [13]. In our research, MPO levels were positively associated with BMI in both HF groups, but not in the control group.

We did not find any difference in circulating levels of MPO between the HFpEF population and healthy individuals. This remains an exciting finding because the previous studies suggest that the MPO levels should rather be elevated in the HFpEF group. However, Kao et al. identified subgroups among HFpEF patients with distinct clinical characteristics and prognoses [26]. It is possible that the HFpEF population in our study belonged to a specific subpopulation that does not have raised circulating levels of inflammatory markers. This finding could potentially be essential information in developing personalized therapies for HFpEF. Standard treatments do not sufficiently target inflammation, which may play a key role in this disease. Recent research showed that an MPO inhibitor-AZD4831 (mitiperstat) decreased inflammation in HFpEF patients, but the efficacy findings were inconclusive [27,28,29]. The Phase 2b-3 clinical trial ENDEAVOR, to investigate the use of the MPO inhibitor AZD483, is ongoing [30]. In the HFrEF group, there were significantly more cases of patients with CAD, a history of MI, and a history of percutaneous coronary intervention (PCI). This is not surprising, as these factors are fundamental to HFrEF etiology. Additionally, these patients had a higher incidence of implanted implantable cardioverter-defibrillator (ICD) or pacemakers, which results from guidelines about primary prevention of sudden cardiac death. Furthermore, this group exhibited a greater prevalence of documented ventricular fibrillation/ventricular tachycardia (VF/VT), which is also associated with MI and reperfusion following PCI.

Our study has several limitations. First, we analyzed levels of markers in peripheral blood rather than directly from the heart. Therefore, the results may be influenced by the fact that these markers could originate from other compartments. However, it is known that HFpEF is a result of the overlap of coexisting extracardiac conditions, and measured nitrosative/oxidative stress may affect the entire body. In general, measuring MPO and MPO-derived products remains expensive and time-consuming. These products are usually found in low concentrations that affect the measurement accuracy. Recently, ELISA kits have been more commonly used, making testing more affordable [8]. Lastly, the study lacked a long-term follow-up of the participants, so we cannot analyze the impact of variables on survival.

## 4. Materials and Methods

This was an observational cohort study conducted at a single cardiac center. A total of 90 patients identified from the hospital medical records were invited to participate in the study. HF was confirmed based on the patient’s symptoms, signs, and transthoracic echocardiography (TTE) results. Preserved ejection fraction was defined as left ventricular ejection fraction (LVEF) ≥ 50%, and reduced ejection fraction was defined as LVEF ≤ 40%. The study population was divided into three groups: HFpEF (n = 27), HFrEF (n = 22), and patients without HF (control, n = 41). The exclusion criteria included heart failure with mid-range ejection fraction (HFmrEF) to precisely differentiate both types of HF (HFpEF and HFrEF). Exclusion criteria also included refusal to participate in the study, an active neoplastic process, an active inflammatory process, a recent history (3 months) of acute MI or cardiac surgery, and new onset and/or exacerbation of any HF. All patients included in the study were in stable condition at the time of enrollment. Comprehensive clinical characteristics and medical records were obtained. An electrocardiogram (ECG) examination was conducted to confirm or rule out the presence of atrial fibrillation (AF). Height and weight were measured, and BMI was calculated. The New York Heart Association (NYHA) functional class for each patient was determined. Based on the echocardiography results, biomarker levels, and clinical assessment, the scores of the HFAPEFF and H2FPEF scales were quantified [31,32].

### 4.1. Blood Sample Collection and Biomarkers Assessment

Peripheral venous blood was collected from each patient from the cephalic vein into citrate tubes. The samples were centrifuged and stored at –80 °C until laboratory assay, not exceeding a storage duration of 6 months. The plasma concentrations of the biomarkers were assessed with the 3-Nitrotyrosine ELISA Kit (Colorimetric) [Novusbio, Littleton, CO, USA, NBP2-66363] and the Myeloperoxidase ELISA Kit (Colorimetric) [Novusbio, NBP2-60581], strictly according to instructions provided by the manufacturer. MPO antibody was used to quantitatively determine autoimmune response to the target antigen. The eGFR was calculated utilizing the CKD-EPI equation based on plasma creatinine levels. Renal dysfunction was characterized by an eGFR of less than 60 mL/min/1.73 m². Anemia was identified by Hgb concentrations below 12 g/dL in females and below 13 g/dL in males.

### 4.2. Echocardiography

TTE was performed by experienced cardiac sonographers using a Philips Affinity 70 Ultrasound machine. All procedures were conducted in accordance with European Society of Cardiology (ESC) guidelines. LV diameters and wall thickness were measured using a parasternal long-axis view. LVEF was determined using Simpson’s method by measuring LV volume in systole and diastole obtained from apical four- and two-chamber views. LV mass index (LVMI) was calculated using the formula for estimation of LV mass from LV linear dimensions and indexed to body surface area (BSA). LV geometry was classified based on relative wall thickness (RWT). LA volume was assessed using the biplane area–length method from apical 2- and 4-chamber views at end-diastole from the frame preceding mitral valve opening and was indexed to BSA (LA volume index, LAVI). The early diastolic peak flow velocity (E velocity) and late diastolic peak flow velocity (A velocity) were assessed through pulsed wave Doppler from the apical 4-chamber view by positioning the sample volume at the tip of the mitral leaflets. Peak early diastolic tissue velocity (e’) and peak systolic tissue velocity (s’) were measured from the septal and lateral aspects of the mitral annulus. The E/e’ ratio was calculated as E velocity divided by mean e’ velocity (average value of the lateral and the septal velocity). The inferior vena cava (IVC) diameter was measured using M-mode echocardiography in subcostal view at end-expiration (IVCmax) and at inspiration (IVCmin). The IVC collapsibility (IVCC) was calculated as (IVCmax − IVCmin)/IVCmax and was expressed as a ratio. Sample echocardiographic measurements from selected patients are presented in Figure 3.

### 4.3. Statistical Analysis

Categorical variables are expressed as numbers and percentages and compared using the χ^2^ test. Quantitative variables are presented as means (standard deviation [SD]) or medians (interquartile range [IQR]). The ANOVA test (normal distribution) and the Kruskal–Wallis H test (non-normal distribution) were performed to compare variables between the three groups. Appropriate post hoc tests were performed (Dunn and Tukey tests). The Shapiro–Wilk test was used to assess the normality of data. The equality of variances was assessed using the Levene test. Statistical data were considered significant with a *p*-value < 0.05. All statistical analyses were performed using RStudio software (version 2022.12.0, Posit Software). Depending on whether the distribution of the quantitative data was normal or not, either the Pearson correlation method or the Spearman analysis was employed to assess linear correlation.

## 5. Conclusions

Nitrosative/oxidative stress measured via circulating levels of 3-NT is significantly intensified in HFpEF but not in HFrEF. The extent of inflammation assessed by MPO concentration is comparable between patients with two types of HF and the healthy population. These findings provide input to the current discussion about the pathogenesis of HF, diagnostic markers, and new therapeutic targets.

## Figures and Tables

**Figure 1 ijms-24-15944-f001:**
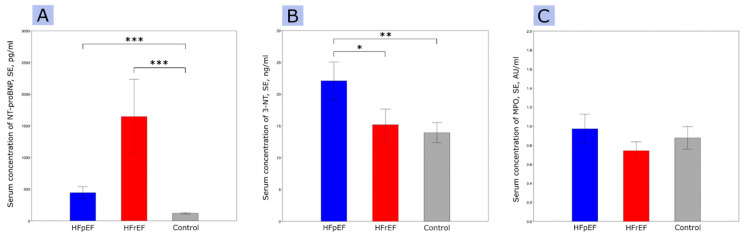
NT-pro-BNP levels (**A**), 3-NT circulating levels (**B**), MPO circulating levels (**C**); abbreviations: 3-NT, 3-nitrotyrosine; HFpEF, heart failure with preserved ejection fraction; HFrEF, heart failure with reduced ejection fraction; MPO, myeloperoxidase; SE, standard error. * *p* < 0.05, ** *p* < 0.01, *** *p* < 0.001.

**Figure 2 ijms-24-15944-f002:**
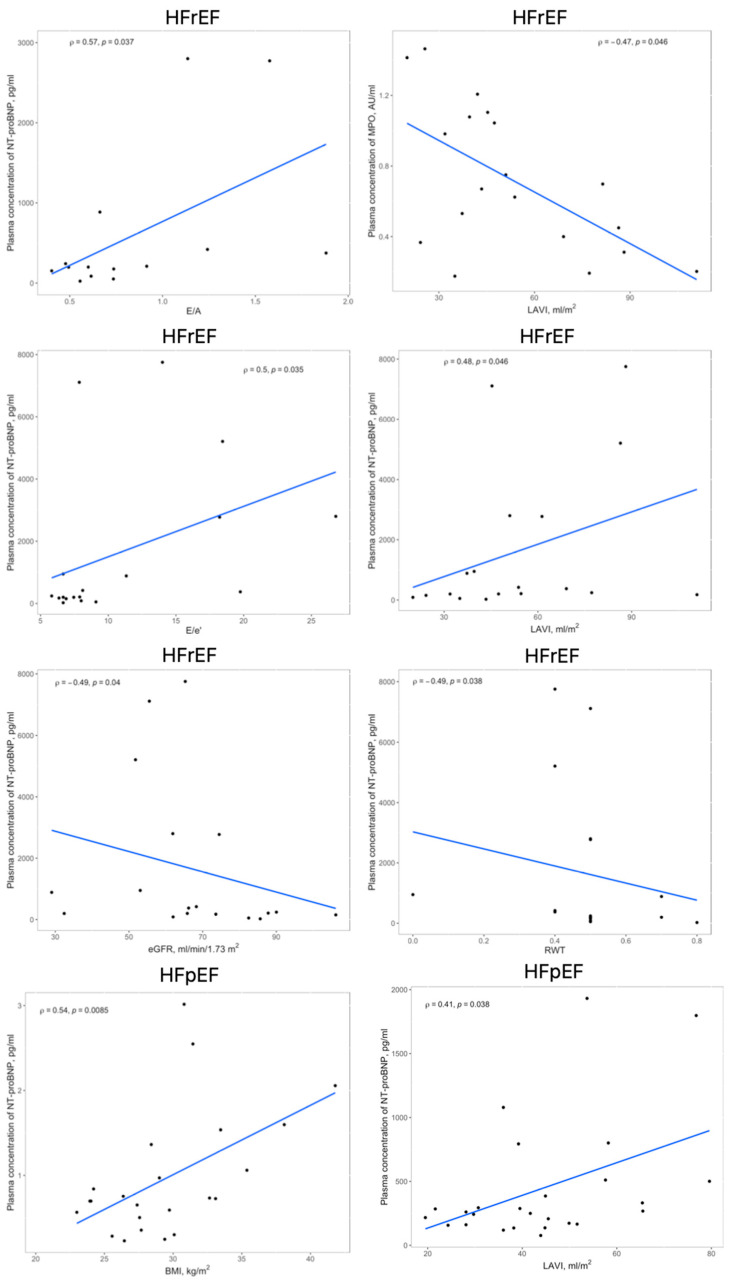
Significant correlations within the HFrEF and HFpEF groups between circulating levels of MPO, 3-NT, or NT-proBNP and echocardiographic findings, BMI or eGFR; abbreviations: 3-NT, 3-nitrotyrosine; BMI, body mass index; eGFR, estimated glomerular filtration rate; HFpEF, heart failure with preserved ejection fraction; HFrEF, heart failure with reduced ejection fraction; LAVI, left atrial volume index; MPO, myeloperoxidase; RWT, relative wall thickness.

**Figure 3 ijms-24-15944-f003:**
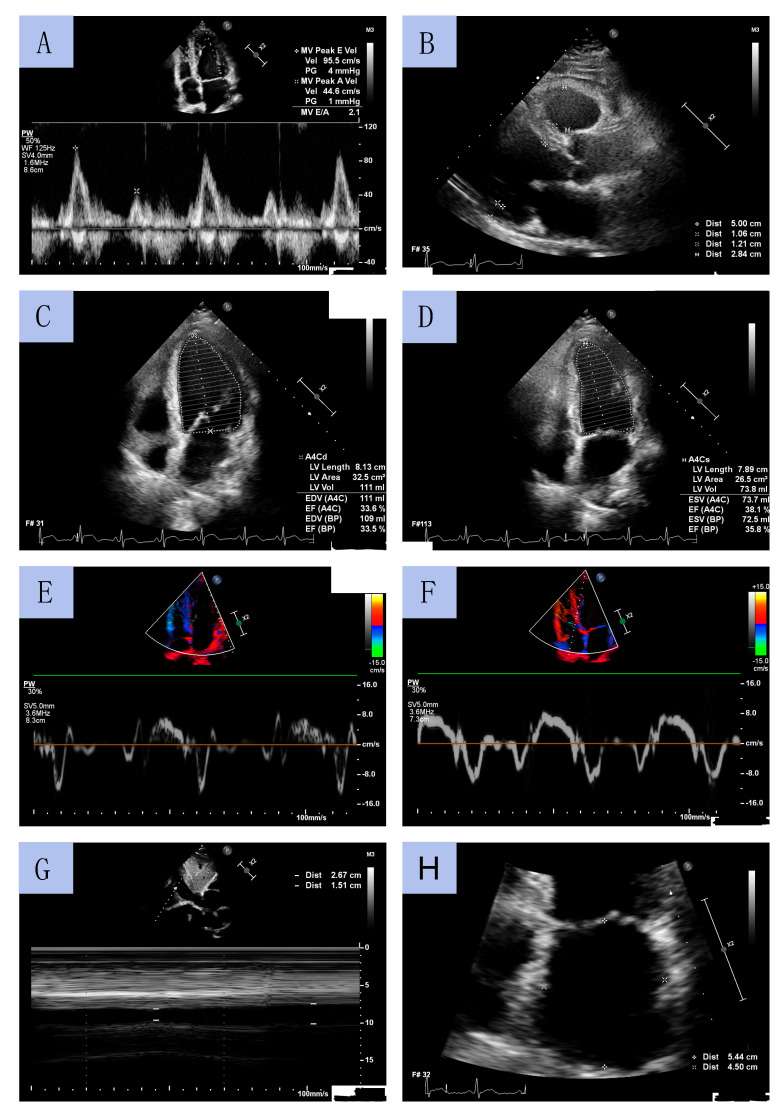
Sample echocardiographic measurements: mitral diastolic peak flow velocities by pulsed wave Doppler (**A**); left ventricle diameters in long axis parasternal view (**B**); left ventricle geometry on end-diastole and end-systole (**C**,**D**); Tissue Doppler Imaging of tissue velocities of the lateral and septal aspects of the mitral annulus (**E**,**F**); the inferior vena cava collapsibility (**G**); left atrium diameters (**H**).

**Table 1 ijms-24-15944-t001:** Baseline characteristics of patients.

Characteristics	HFpEF (N = 27)	HFrEF (N = 22)	Control (N = 41)	*p*-Value
Male gender, n (%)	15 (56)	16 (73)	15 (37)	0.0203
Age, years, median (IQR)	72.00 (64.00–76.00)	74.50 (61.00–79.00)	70.00 (64.00–73.00)	ns
BMI, kg/m^2^, median (IQR)	29.00 (26.37–32.65)	28.39 (25.39–31.25)	29.32 (26.81–32.47)	ns
NYHA II, n (%)	25 (93)	12 (55)	-	0.0021
NYHA III, n (%)	2 (7)	10 (45)	-	
H2FPEF score, median (IQR)	5 (4–6)	4.5 (3–6)	5 (3–6)	ns
HFA-PEFF score, median (IQR)	4 (4–6) ^1^	5 (4–6) ^2^	3 (2–4)	<0.0001
Medical History				
Hypertension, n (%)	23 (85)	15 (68)	33 (80)	ns
Diabetes or prediabetes, n (%)	4 (15)	11 (50)	8 (20)	0.0094
Lipid disorders, n (%)	4 (15)	2 (9)	13 (32)	ns
Renal dysfunction, n (%)	4 (15)	7 (32)	6 (15)	ns
Anemia, n (%)	4 (15)	5 (23)	6 (15)	ns
Obesity, n (%)	11 (41)	9 (41)	18 (44)	ns
Thyroid disease, n (%)	5 (19)	5 (23)	16 (39)	ns
Chronic obstructive pulmonary disease, n (%)	1 (4)	1 (5)	0 (0)	ns
Currently smoking, n (%)	2 (7)	3 (14)	1 (2)	ns
CAD, n (%)	16 (59)	16 (73)	9 (22)	0.0001
History of MI, n (%)	6 (22)	10 (45)	3 (7)	0.0018
History of PCI, n (%)	8 (30)	9 (41)	3 (7)	0.0051
History of CABG, n (%)	2 (7)	3 (14)	2 (5)	ns
Cardiomyopathy	0 (0)	6 (28)	2 (5)	0.0058
History of AF, n (%)	19 (70)	13 (59)	19 (46)	ns
AF at the time of enrollment, n (%)	4 (15)	5 (23)	1 (4)	0.0380
Implanted ICD/pacemaker, n (%)	2 (7)	16 (73)	4 (10)	<0.0001
Documented VF/VT, n (%)	0 (0)	4 (18)	2 (5)	0.0329
Selected Medications				
Diuretics	23 (85)	15 (68)	33 (80)	ns
ACEI/ARB	24 (89)	18 (82)	37 (90)	ns
Beta-blocker	24 (89)	19 (86)	34 (83)	ns
SGLT2i	5 (19)	20 (91)	11 (27)	<0.0001
Statin	17 (63)	17 (77)	19 (46)	ns
NOAC	17 (63)	13 (59)	10 (24)	0.0021

^1^ HFpEF vs. control, *p* < 0.001, ^2^ HFrEF vs. control, *p* < 0.001. Abbreviations: ACEI, angiotensin-converting enzyme inhibitor; AF, atrial fibrillation; ARB, angiotensin-receptor blocker; BMI, body mass index; CABG, coronary artery bypass grafting; CAD, coronary artery disease; eGFR, estimated glomerular filtration rate; HFpEF, heart failure with preserved ejection fraction; HFrEF, heart failure with reduced ejection fraction; ICD, implantable cardioverter-defibrillator; IQR, interquartile range; MI, myocardial infarction; NOAC, non-vitamin K antagonist oral anticoagulants; ns, non-significant; NYHA, New York Heart Association; PCI, percutaneous coronary intervention; SGLT2i, sodium-glucose cotransporter-2 inhibitor; VF, ventricular fibrillation; VT, ventricular tachycardia.

**Table 2 ijms-24-15944-t002:** Echocardiographic characteristics.

Characteristics	HFpEF (N = 27)	HFrEF (N = 22)	Control (N = 41)	*p*-Value
LVEF, %, median (IQR)	55 (50–60) ^1^	37.5 (30–40) ^2^	55 (50–60)	<0.0001
E velocity, cm/s, mean (SE)	76.15 (4.57)	62.36 (4.43)	69.45 (2.21)	ns
A velocity, cm/s, mean (SE)	70.62 (3.87)	75.41 (4.15)	75.90 (3.36)	ns
s’ septal, cm/s, median (IQR)	6.5 (6–8)	5 (4–6.75) ^2^	8 (7–9)	<0.0001
e’ septal, cm/s, median (IQR)	7 (5–8)	4 (3.75–6.25) ^2^	7 (6–8)	0.0023
s’ lateral, cm/s, median (IQR)	7.5 (6–8.25) ^3^	5.5 (4–8) ^2^	8 (7–9)	0.0067
e’ lateral, cm/s, median (IQR)	9.5 (7–11) ^1^	6 (5–9.25)	8 (6–11)	0.0309
E/A, ratio, median (IQR)	1.17 (0.76–1.59)	0.74 (0.58–1.30)	0.91 (0.80–1.17)	ns
E/e’, ratio, median (IQR)	9.54 (7.69–12.22)	8.06 (6.67–18.00)	8.88 (7.30–11.34)	ns
IVS, cm median (IQR)	1.20 (1.10–1.33)	1.35 (1.12–1.5) ^2^	1.2 (1.05–1.30)	0.0472
LVEDD, cm median (IQR)	4.65 (4.25–5.03) ^1^	5.63 (4.98–6.05) ^2^	4.6 (4.25–5.00)	<0.0001
PW, cm median (IQR)	1.1 (0.90–1.33) ^1^	1.3 (1.2–1.50) ^2^	1.1 (0.95–1.30)	0.0015
LAVI, mL/m^2^, mean (SE)	44.23 (3.14)	53.60 (5.25) ^2^	35.53 (1.62)	0.0003
LVMI, g/m^2^, median (IQR)	103.10 (84.10–110.60) ^1^	192.35 (145.50–222.00) ^2^	97.50 (80.70–116.50)	<0.0001
RWT, median (IQR)	0.50 (0.40–0.60)	0.50 (0.40–0.50)	0.50 (0.40–0.60)	ns
IVCC, %, mean (SE)	45 (3)	41 (4)	49 (4)	ns
Moderate valve disease, n (%)	3 (11)	1 (5)	3 (7)	ns
Severe valve disease, n (%)	3 (11)	6 (27)	0 (0)	0.0026

^1^ HFpEF vs. HFrEF, *p* < 0.001, ^2^ HFrEF vs. control, *p* < 0.001, ^3^ HFpEF vs. control. Abbreviations: HFpEF, heart failure with preserved ejection fraction; HFrEF, heart failure with reduced ejection fraction; IQR, interquartile range; IVCC, inferior vena cava collapsibility; IVS, intraventricular septum; LAVI, left atrial volume index; LVEDD, left ventricle end-diastolic diameter; LVEF, left ventricular ejection fraction; LVMI, left ventricular mass index; ns, non-significant; PW, posterior wall; RWT, relative wall thickness; SE, standard error.

## Data Availability

This study’s data cannot be shared publicly for privacy reasons. The data will be shared upon request to the corresponding author.

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
