# Peer review of "Evaluation of Nitrosative/Oxidative Stress and Inflammation in Heart Failure with Preserved and Reduced Ejection Fraction"

_ijms, 2023, doi:10.3390/ijms242115944_

Round 1
Reviewer 1 Report
Comments and Suggestions for Authors
The authors have discussed a very interesting topic in this article related to the differences in nitrosative/oxidative stress and inflammation heart failure parameters in HFrEF and HFpEF but there are a few concerns that need to be addressed. Some of them are mentioned below.
1. Add references in the introduction section where the effect of oxidative stress is mentioned with respect to hypertension and explain the mechanism in a bit more detail.
a. https://doi.org/10.1002/advs.202303259
2. The authors must provide the raw ECG data for a few of the selected patients as proof for HFrEF and HFpEF based on their claims. The raw data must be from Pulse wave doppler and tissue doppler sections as they are going give in-depth volumetric data as well as well as any change in the ventricular masses. The data from M-mode echo also needs to be added to show changes in fractional shortening in the left ventricle. The tabular data must be accompanied by pictorial data from ECG.
3. Although authors have mentioned ELISA to be a more robust and cheaper technique, ELISA is not considered conclusive results in these kinds of studies and the authors are requested to show data through quantification of biomarkers such as for MPO and 3-NT levels through specific related gene expressions to support the claims.
Comments on the Quality of English LanguageMinor paraphrasing of some sentences is needed to make them easier for the readers.
Author Response
Dear Reviewer,
Thank you for your valuable comments and suggestions. Below are the changes we have made:
1: We have incorporated new references and provided a more detailed explanation of the oxidative stress mechanism in hypertension. Changes have been highlighted in yellow. (Line 61-70)
2: We have included a figure with sample echocardiographic measurements in the manuscript from selected patients. (Line 122-123)
3: We agree with your insight. Your recommendation to quantify biomarkers such as MPO and 3-NT levels through specific gene expressions is indeed an excellent idea to enhance the comprehensiveness of our study.
We must acknowledge that this study was preliminary in nature, and we operated under specific constraints, including a limited budget, which led us to utilize ELISA tests. However, we fully recognize the potential of exploring gene expression for these molecules in future research, as it would significantly contribute to the depth and scope of this project.
4: We have made minor paraphrasing adjustments to some sentences to enhance readability for our readers. Changes have been highlighted in yellow.
Thank you for your feedback.
Sincerely,
Corresponding author

Reviewer 2 Report
Comments and Suggestions for Authors
1. Line 61-63. Under oxidative stress, human endothelial cells in the endocardium express MPO, which consumes NO and converts it to a nitrogen radical, causing protein nitration and tissue damage. This may be one of the causes of HF [15].
Note. The cause of HF may be previous MI, arterial hypertension, or atrial fibrillation - but not oxidative stress.
2. Line 74-79. Exclusion criteria also included - a recent history (3 months) of acute myocardial infarction… All patients included in the study were in stable condition for at least 3 months.
Note. Patients who had an MI 3 months ago cannot be stable for the last 3 months, please specify by statement.
3. Patients with HF, at least with reduced EF, did not receive b-blockers, ACE inhibitors, MRA, or diuretics. Why did patients not receive standard therapy, and how does this relate to modern recommendations for the treatment of heart failure?
4. 8 patients (20%) in the control group have Diabetes or prediabetes, but 10 patients (24%) receive SGLT2i. What are the indications for patients without Diabetes or prediabetes?
5. A lot of TTE is indicated in section 2.2 Echocardiography, but only a few in the table. Review the required list and reduce or present the results.
Author Response
Dear Reviewer,
Thank you for your valuable comments and suggestions. Here are the changes we have made in response to your feedback:
1: We have revised this misleading sentence. The second reviewer also pointed out this section and provided suggestions. Consequently, we have made substantial improvements to this part of the text. Changes have been highlighted in yellow. (Line 61-70)
2: We understand your concerns, and we have revised this section of the text to clarify that patients included in the study were in a stable condition at the time of enrollment. Patients who had experienced a myocardial infarction within the last 3 months were excluded from the study. Changes have been highlighted in yellow. (Line 79-82)
3: We would like to emphasize that the listed medications were not an exhaustive representation of all the drugs the patients took, as we believed that the content in the table would become excessively detailed.
We selected these medications based on established findings from studies that indicate statins, NOACs, or SGLT2i might affect pathways associated with nitric oxide and nitrosative stress.
It is essential to underline that the patients were treated following the then-current ESC guidelines, which did not yet consider SGLT2 inhibitors for HFpEF. Additionally, some patients had suboptimal compliance, discovered during medical interviews. However, we added several other medications to Table 1 to demonstrate that, in general, patients received standard treatment. Changes have been highlighted in yellow.
4: In this case, SGLT2 inhibitors were used in patients with diagnosed impaired renal function, even in cases where diabetes or prediabetes were not present. https://doi.org/10.1093/ckj/sfaa198
5: Indeed, the section on methodology and transthoracic echocardiography (TTE) is quite extensive. We aimed to provide a comprehensive overview of the various stages of this examination to ensure clarity for the readers. However, we understand your point about the volume of data in the table. Thus, we incorporated the specific missing parameters in Table 2 to provide a more comprehensive overview. Changes have been highlighted in yellow.
Thank you for your feedback.
Sincerely,
Corresponding author.

Round 2
Reviewer 1 Report
Comments and Suggestions for Authors
I would like to congratulate the authors for their valuable insights and for improving the contents of the manuscript as directed by reviewers. I would like to see the next version of the research work where the authors have promised to show the gene expression studies as their validation and proof serving towards a significant transition in this field of research.
Reviewer 2 Report
Comments and Suggestions for Authors
No.